# Mechanochemically Synthesised Coal-Based Magnetic Carbon Composites for Removing As(V) and Cd(II) from Aqueous Solutions

**DOI:** 10.3390/nano9010100

**Published:** 2019-01-16

**Authors:** Anton Zubrik, Marek Matik, Michal Lovás, Zuzana Danková, Mária Kaňuchová, Slavomír Hredzák, Jaroslav Briančin, Vladimír Šepelák

**Affiliations:** 1Institute of Geotechnics, Slovak Academy of Sciences, Watsonova 45, SK-04001 Košice, Slovakia; matik@saske.sk (M.M.); lovasm@saske.sk (M.L.); orolinova@saske.sk (Z.D.); hredzak@saske.sk (S.H.); briancin@saske.sk (J.B.); vlsep@saske.sk (V.Š.); 2Faculty of Mining, Ecology, Process Control and Geotechnologies, Technical University of Košice, Letná 9, SK-04200 Košice, Slovakia; maria.kanuchova@tuke.sk; 3Institute of Nanotechnology, Karlsruhe Institute of Technology, Hermann-von-Helmholtz-Platz 1, D-76344 Eggenstein-Leopoldshafen, Germany

**Keywords:** magnetic carbon, mechanochemical synthesis, coal, sorption, arsenic

## Abstract

The continued decrease in water quality requires new advances in the treatment of wastewater, including the preparation of novel, effective, environmentally friendly, and affordable sorbents of toxic pollutants. We introduce a simple non-conventional mechanochemical synthesis of magnetically responsive materials. Magnetic lignite and magnetic char were prepared by high-energy ball co-milling from either raw Slovak lignite or coal-based char together with a ferrofluid. The products were characterised by X-ray diffraction, electron microscopy, ^57^Fe Mössbauer spectroscopy, X-ray photoelectron spectroscopy (XPS), volumetric magnetic susceptibility, and low-temperature nitrogen adsorption, and both magnetic carbons were comparatively tested as potential sorbents of As(V) oxyanions and Cd(II) cations in aqueous solutions. The magnetic char was an excellent sorbent of As(V) oxyanions (*Q*_m_ = 19.9 mg/g at pH 3.9), whereas the magnetic lignite was less effective. The different sorption properties towards arsenic anions may have been due to different oxidation states of iron on the surfaces of the two magnetic composites (determined by XPS), although the overall state of iron monitored by Mössbauer spectroscopy was similar for both samples. Both magnetic composites were effective sorbents for removing Cd(II) cations (*Q*_m_ (magnetic lignite) = 70.4 mg/g at pH 6.5; *Q*_m_ (magnetic char) = 58.8 mg/g at pH 6.8).

## 1. Introduction

High-quality water is essential for living organisms. Rapid industrialisation has produced hazardous secondary products on a large scale, polluting water especially in developing countries. Several methods including ozonation, biodegradation, electrochemical oxidation, and adsorption have been established for eliminating toxic substances either in organic (e.g., pharmaceuticals, personal-care products, and reactive dyes) or inorganic (heavy metals) forms. Each method has advantages and disadvantages. The development of new efficient techniques, further improvements and modifications of existing methods, or combining treatment processes is therefore necessary. Sorption is one of the best and most commonly used procedures in the treatment of wastewater [1,2]. New environmentally friendly and regenerable materials are still being developed for improving sorption properties and minimizing the cost. For example, a new mussel-inspired polydopamine microsphere was prepared by in situ oxidative polymerisation for rapid Cr(VI) removal [3]. A novel graphene-based sponge for removing crude oil was investigated by Shiu et al. [4]. A multi-walled carbon-nanotube nanocomposite has been used for the selective removal of copper ions from contaminated water [5]. Other types of novel adsorbents can be synthesised from natural sources such as lignocellulosic material (e.g., agricultural waste biomass, coal), fermentation products, and microbial biomass [6,7,8,9]. Adsorption on carbonaceous materials offers low-energy consumption, lack of by-products, simplicity, reliability, and low cost. These materials are very effective, but separating them easily and effectively from wastewater in a continuous-flow system remains a challenge [10]. The modification of adsorbents by magnetic particles is therefore one possible way of removing them. Magnetic adsorbents are attractive materials for wastewater treatment, because they can be easily removed together with adsorbed pollutants from aqueous environments by an external magnetic field [10,11,12]. Modification of carbon matrices with iron-bearing particles may also increase sensitivity to oxyanions [11,13]. Many available magnetic materials/composites have great potential for removing either inorganic pollutants such as toxic metals in cationic (Cd, Cu, Zn, Hg, Pb etc.) and anionic (As, Cr) forms or organic pollutants (e.g., polycyclic aromatic hydrocarbons, pesticides, organic dyes, and pharmaceuticals). Iron-based magnetic sorbents are very beneficial for removing highly toxic arsenic oxyanionic species, which are one of the major environmental health hazards in contaminated groundwater [10,11,12]. Magnetic adsorbents can be divided into three main groups based on the type of material: (i)pure synthetic inorganic compounds/oxides of metals such as Fe, Co, Ni, and Cu (e.g., Fe_3_O_4_, α-Fe_2_O_3_, γ-Fe_2_O_3_, β-FeOOH, and CuFe_2_O_4_),(ii)magnetic silica-based composites (Si-containing polymers such as zeolites and natural or synthetic aluminosilicate compounds used as matrices modified/treated by magnetic particles, mainly iron oxides), and(iii)magnetic carbon composites (carbon matrices modified with magnetic particles), e.g., activated carbon, biochars, fullerenes, nanotubes, and other carbon materials used as matrices.

Various methods have been introduced for preparing low-cost magnetic carbon composites. Co-precipitation is one of the most commonly used methods for their synthesis, where metal cations are precipitated to metal oxides and then loaded on the carbon matrix [14]. Magnetic composites can also be synthesised directly during pyrolysis. The composites are prepared by co-pyrolysing a carbonaceous material (plant-biomass waste, coal, microbial biomass, or organic-based industrial products, e.g., tires and petroleum products) mainly with iron oxides. For example, Wang et al. [15] synthesised magnetic carbon composites by pyrolysing a mixture of hematite and pinewood biomass.

Several enhancements to the conventional methods of synthesis can be applied to improve the sorption properties of magnetic carbon composites (e.g., using microwave heating for synthesising magnetic composites [11], ultrasound irradiation to improve the distribution of magnetic particles in the matrix, hydrothermal carbonisation, spray pyrolysis, and microwave synthesis of magnetic particles for modifying non-magnetic materials [16]) Novel approaches, however, are also available for synthesising magnetic carbon composites. Maya et al. [17] described the preparation of magnetic porous carbon from metal-organic frameworks for analytical chemical application in solid-phase extraction. The non-conventional mechanochemical approach can also be used for synthesising magnetically responsive materials. Safarik et al. [18], for example, described the mechanochemical synthesis of magnetic carbon from non-magnetic precursors. The mechanochemical approach has several advantages, e.g., simplified processes to decrease the cost (fewer processing steps), ecological safety, and large-scale production of nanosized crystallites with unique properties from the mechanochemical synthesis of nanomaterials [19,20]. Mechanochemistry can also be applied in many fields of material sciences (e.g., extractive metallurgy, crystal engineering, materials engineering, coal industry, building industry, agriculture, pharmacy, and waste treatment) [21]. The mechanical activation of coal affects the organic structure of the coal [22,23]. The mechanical deformation of the organic structure by activation grinding has been associated with new surface areas, solid-state defects, amorphisation, and metastability, which greatly affect and substantially increase chemical reactivity. 

We applied a non-conventional mechanochemical approach for synthesising magnetically responsive materials from Slovak coal and a magnetic fluid. Native coal and coal char were used as carbon matrices, and ferrofluid (FF) was used as a source of magnetic properties. Magnetic carbons were characterised by X-ray diffraction, low-temperature nitrogen adsorption, Mössbauer spectroscopy, X-ray photoelectron spectroscopy (XPS), and electron microscopy. The prepared magnetic composites were subsequently tested as sorbents of As(V) oxyanions and Cd(II) cations in aqueous solutions. Note that cadmium and arsenic are highly toxic pollutants. The US Environmental Protection Agency has set the maximum contaminant levels for cadmium and arsenic in drinking water at 0.005 and 0.01 mg/L, respectively. Both metals cause serious medical complications such as cancer. 

## 2. Materials and Methods

### 2.1. Preparation of Magnetic Carbons

A representative sample of Slovak lignite from Čáry Colliery (48°39′34.524″ N, 17°4′32.574″ E) was used. Biochar was prepared by pyrolysing lignite in a horizontal quartz furnace in a nitrogen atmosphere at 550 °C (heating rate, 12 °C/min; holding time of pyrolysis, 135 min). FF containing maghemite nanoparticles [11] was prepared following a patented procedure [24]. Briefly, water-based FF was prepared by precipitating Fe^2+^ and Fe^3+^ salts with ammonia at 80 °C in a water bath. Magnetic particles were then washed with deionised water to neutral pH and covered with oleic acid, and a suitable surfactant was used to disperse the particles in a carrier fluid (e.g., water or kerosene).

Magnetic carbon was synthesised either from raw lignite or biochar as follows: 10 g of lignite or biochar were mixed with 20 mL of FF with a volumetric magnetic susceptibility of 284,000 × 10^−6^ SI units. The concentration of iron in FF was 4% (measured by atomic absorption spectroscopy, see Section 2.7). The mixtures (lignite/FF and biochar/FF, respectively) were prepared for synthesis by drying at 85 °C for 8 h. Magnetic carbon was synthesised by high-energy ball milling using a planetary Pulverisette 6 mill (Fritsch GmbH, Idar-Oberstein, Germany) in a nitrogen atmosphere under the following conditions: initial granularity of the sample, <0.2 mm; grinding speed, 550 rpm; grinding time, 180 min; and sample weight, 10 g. The grinding chamber (250 mL) and balls (5 mm in diameter) were made of tungsten carbide. The ball:powder weight ratio was 40:1.

### 2.2. Volumetric Magnetic Susceptibility

Volumetric magnetic susceptibility (κ) was measured by a Kappabridge KLY-2 apparatus (Geophysics, Brno, Czech Republic) under the following conditions: magnetic field intensity, 300 A/m; field homogeneity, 0.2%; and frequency, 920 Hz.

### 2.3. X-ray Powder Diffraction (XRD)

XRD was studied using a D8 Advance diffractometer (Bruker AXS, Karlsruhe, Germany) with Cu Kα radiation (voltage, 40 kV; current, 40 mA; goniometer step, 0.05°/s; and time step, 15 s).

### 2.4. Textural Analysis

Surface properties of the samples were determined from the adsorption and desorption isotherms measured using a NOVA 1200e Surface Area & Pore Size Analyzer (Quantachrome Instruments, Boynton Beach, FL, USA) by the physical adsorption of nitrogen at −196 °C. The samples were degassed at 100 °C in a vacuum oven at a pressure <2 Pa for 16 h prior to the measurements. The measurement data were processed using the Brunauer–Emmett–Teller (BET) isotherm [25] in a range of relative pressure of 0.05–0.3 to obtain the specific surface area (S_A_). The external surface (S_ext_) and volume of micropores (V_micro_) were calculated from a t-plot using the Harkins-Jura standard isotherm. Total pore volume (V_tot_) was estimated from the maximum adsorption at a relative pressure similar to the saturation pressure. The distribution of pore sizes was obtained from the desorption isotherm using the Barrett-Joyner-Halenda method [26].

### 2.5. Mössbauer Spectroscopy

^57^Fe Mössbauer spectra were recorded with 512 channels and measured at room temperature using a laboratory Mössbauer spectrometer in constant-acceleration mode equipped with a ^57^Co(Rh) source. The Mossbauer spectra were processed (i.e., noise filtering and fitting) using MossWinn program. Isomer shifts were referred to an α-Fe foil sample at room temperature. 

### 2.6. Morphology

Particle morphology was studied by field-emission scanning electron microscopy (FE-SEM) TESCAN MIRA3 FE (TESCAN, Brno, Czech Republic). 

### 2.7. Elemental Analysis and Ash Content

Elemental carbon, hydrogen, nitrogen, and sulphur (CHNS) were analysed using a Vario MACRO cube elemental analyser (Elementar Analysensysteme GmbH, Hanau, Germany) equipped with a thermal-conductivity detector. The combustion and reduction tubes were set to 1150 and 850 °C, respectively. Sulphanilamide (41.81% C, 16.26% N, 4.65% H, and 18.62% S) was used as a CHNS standard.

Ash content was determined by ignition in a muffle oven at 815 °C to a constant weight (after 3 h). Aqua regia (a mixture of nitric acid and hydrochloric acid at a molar ratio of 1:3) was used to dissolve the magnetic carbon biochars and FF for determining the total iron content by atomic absorption spectroscopy (AAS; Varian 240 RS/240 Z, Mulgrave, Australia).

### 2.8. XPS

XPS was measured using an XPS instrument (SPECS GmbH, Berlin, Germany) equipped with a PHOIBOS 100 SCD and a non-monochromatic X-ray source. The survey surface spectrum was measured at a transition energy of 70 eV, and core spectra were measured at 20 eV at room temperature. All spectra were acquired at a basic pressure of 2 × 10^−8^ mbar with AlK_α_ excitation at 10 kV (150 W). The data were analysed by SpecsLab2 CasaXPS (Casa Software Ltd., Teignmouth, UK). A Shirley and Tougaard type baseline was used for all peak fits. The spectrometer was calibrated against silver (Ag 3d). All samples had variable degrees of charging due to their insulating nature. This problem was resolved by calibration to carbon. The XPSs of the Fe 2p spectra were recorded to obtain information on elemental composition.

### 2.9. Zeta Potential (ZP)

ZP was measured using a Zetasizer Nano ZS (Malvern Panalytical Ltd., Malvern, UK). The Zetasizer Nano measures the electrophoretic mobility of particles. Malvern Zetasizer software was used for evaluation. The data for electrophoretic mobility data were converted into the zeta potential using Henry’s equation and Smoluchowski approximation. Smoluchovski approximation is valid for polar media and particles >200 nm. Particle size of our composites was in the range of 300–700 nm (measured by photon cross-correlation spectroscopy using a Nanophox particle-size analyser (Sympatec GmbH, Clausthal-Zellerfeld, Germany)). Sample ZP (concentration of 2 g/L) was measured in 0.1 M NaNO_3_ within various pH ranges, which were adjusted by the addition of 2 M NaOH or HNO_3_. The measurements were repeated three times for each sample.

### 2.10. Sorption Experiments

The sorption properties of the magnetic carbons were studied for As(V) and Cd(II) under batch-type conditions. The sorbent concentration was 2 g/L. The experiments were performed in a rotary shaker at 30 rpm at room temperature, and the equilibrium time was 24 h. Model solutions of As(V) were prepared by dissolving AsHNa_2_O_4_·7H_2_O in deionised water. A model cadmium solution was prepared by dissolving Cd(NO_3_)_2_·4H_2_O in deionised water. The initial concentrations (C_0_) of As(V) and Cd(II) were 98.6 and 92 mg/L, respectively, for determining the influence of pH. Various initial concentrations were used for determining adsorption isotherms (C_0_(As) = 10–1000 mg/L; C_0_(Cd) = 10–464 mg/L). Metal quantity (As, Cd, and Fe) in the solutions was determined by AAS. The sorption experiments were evaluated by the Langmuir model [27] (an isotherm for monolayer adsorption on a homogenous surface) and the Freundlich model [28] (an isotherm for multilayer adsorption on a heterogenous surface). Adsorption was thus well described by constants obtained from the two models. 

The Langmuir isotherm is defined by:(1)$$q_{e}=Q_{m}\frac{bC_{e}}{1+bC_{e}}$$
where *q_e_* is equilibrium adsorption capacity (mg/g), *Q_m_* is the maximum adsorption capacity (mg/g), *C_e_* is the equilibrium metal concentration, and *b* is a Langmuir constant characterising the affinity between the adsorbed molecule and the adsorbent (L/mg).

The Freundlich isotherm is defined by: (2)$$q_{e}=K_{F}C_{e}{}^{\frac{1}{n}}$$
where *K_F_* (L/g) and *n* are constants of the isotherm.

*Q_m_*, *b*, *K_F_*, and *n* were determined from the experimental data using the linearised forms of the above equations.

The linearised form of the Langmuir isotherm is:(3)$$\frac{C_{e}}{q_{e}}=\frac{C_{e}}{Q_{m}^{}}+\frac{1}{Q_{m}b}$$

The most important value is the slope (1/*Q_m_*), which is the inverse of the maximum adsorption capacity.

The linearised form of the Freundlich isotherm is:(4)$$\ln\;q_{e}=\ln\;K_{F}+\frac{1}{n}\ln C_{e}$$

## 3. Results and Discussion

### 3.1. Characterisation of Initial Materials (Coal and Char) and the Synthesised Magnetic Sorbents

A first, basic analysis of Slovak lignite as an initial material and an elementary CHNS analysis of the sorbents were carried out (Table 1). The proximate analysis found that the Slovak lignite contained 13.8% ash, 7.5% moisture, 43.9% volatile compounds, and 42.3% fixed carbon. The chemical analysis of the ash found that the Slovak lignite contained typical inorganic species such as SiO_2_ (2.2%), CaO (2.1%), Al_2_O_3_ (1.9%), Fe_2_O_3_ (1.7%), sulphur (1.4%), Na_2_O (0.9%), and MgO (0.6%). The percentage distribution of other inorganic elements was <0.1%.

Table 1 shows the results of the elemental analysis of the initial material (lignite and char) and the synthesised magnetic sorbents. The carbon and hydrogen contents typically differed between the native lignite and char (pyrolysed lignite at 550 °C). The char had much more carbon (62.9%) than the lignite (51.9%), but the hydrogen content was lower: 3.71% (pyrolysed coal, i.e., char) compared to 5.4% (native coal). The carbon content decreased in the samples of magnetic sorbents after the coal/char was mixed with FF and the subsequent mechanical treatment. Iron content was similar in the two magnetic sorbents (7.7% for magnetic lignite and 8.4% for magnetic char). The magnetic properties of the magnetic sorbents were also compared. They were evaluated by determining *κ* measured in a constant magnetic field. Both char and lignite are paramagnetic materials with relatively low *κ* (κ_char_ = 233 × 10^−6^ SI units; κ_lignite_ = 89 × 10^−6^ SI units). The composites acted as strong magnetic materials after mechanosynthesis. κ for the char and lignite samples increased 1420- and 3860-fold, respectively, due to the occurrence of iron-bearing nanocomponents (e.g., nanomagnetite/nanomaghemite) in the pores of the carbon matrices. Interestingly, the magnetic susceptibility was higher for the magnetic lignite than the magnetic char, even though the magnetic lignite contained less iron than the magnetic char. As mentioned above, magnetic susceptibility predicts the ability of the synthesised magnetic sorbents to remove sorbents from aqueous solutions for application in filtered systems.

### 3.2. Surface Analysis of the Magnetic Sorbents

High-energy ball milling, a non-conventional mechanochemical process, was selected for synthesising the magnetic carbons to produce homogenous materials with a high *κ* and suitable textural properties. The morphology of the two samples (magnetic lignite and magnetic char) observed by FE-SEM was similar and characterised by spherical structures on their surfaces (Figure 1). These structures with different sizes (from tens to hundreds of nanometres) probably originated from the spherical FF particles and were homogenously distributed in the carbon matrices. 

The adsorption isotherms were analysed in detail to compare the textural properties of the materials (see Figure 2). The volume of the adsorbed gas increased moderately for both magnetic samples; it changed almost linearly within the wide range of relative pressures and was larger for the magnetic char. A considerable increase in gas volume at a relative pressure (p/p_0_) of 0.9 from the adsorption isotherm indicated the presence of large pores, i.e., macropores. Both magnetic samples had narrow, almost horizontal hysteretic loops, indicating a small presence of mesoporous structure. The magnetic char had more open loop. The so-called forced closure of the hysteretic loop at a relative pressure p/p_0_ of ~0.45 was not observed. This phenomenon could be associated with the specific interaction between the diameter and character of the pores (size and eventually type and charge of surface ions) and nitrogen molecules with their quadrupole effect. 

The distribution curves of pore size for the magnetic sorbents and precursors (char and lignite) are presented in the inset in Figure 2. For example, the content of meso- and macropores differed considerably between the magnetic char and the char precursor. The broad distribution in the range of larger mesopores (>20 nm in diameter) and macropores for the char precursor shifted to smaller mesopores for magnetic char. Their content also increased consistent with a considerable increase in the total pore volume (see the textural parameters for the synthesised sorbents and precursors in Table 2). The distribution curve was similar for the magnetic lignite, with a slight increase in total pore volume. The data derived from the adsorption isotherm were processed by the BET isotherm for calculating the specific surface area. 

The intercepts and C_BET_ constants were negative for both char samples (magnetic char and char). Their specific surface areas therefore did not have a real physical meaning, perhaps due to the predominant presence of macropores. The texture of the samples was evaluated from a t-plot using the Harkins-Jura isotherm, which allows determining the micropore volume and the external surface area. From the obtained values of micropore volume it can be concluded that all studied samples are not microporous. The high-energy ball milling of the lignite and char increased the external surface, particularly for the magnetic char (Table 2). These results corresponded to the shape of the adsorption/desorption isotherms and pore-size distribution curves (see Figure 2).

Figure 3 shows the ZP of the sorbents at various pHs. The magnetic char had a more positive ZP, which is correlated with better properties of sorption to arsenic oxyanions. ZP, however, was nevertheless negative due to the strong negative surface charge of the lignite and char samples. Finally, ZP of the magnetic char was measured in the presence of arsenic anions for comparison. The arsenic anions clearly decreased ZP (to more negative values) of the adsorbent throughout the pH range.

### 3.3. Structural Differences/Similarities of the Prepared Magnetic Sorbents

Figure 4 shows the X-ray diffraction patterns of the dried FF and synthesised sorbents. Several broad peaks were recorded, which could be due to the spinel structure of nanomagnetite or nanomaghemite. Note that X-ray diffraction cannot differentiate between them, especially when the particles are extremely small with poor structural ordering and have a broad size distribution (several nm). We therefore used ^57^Fe Mössbauer spectroscopy for a detailed investigation of iron-phase composition. 

Figure 5 shows the room-temperature ^57^Fe Mössbauer spectrum of the dried FF, magnetic lignite, and magnetic char. The initial sample (dried FF) had a sextet with broad spectral lines (Figure 5c), which were fitted by a distribution of hyperfine magnetic fields. An isomer shift of 0.34 mm/s and quadrupole splitting near zero are typical for Fe^3+^ cations with a cubic symmetry in the spinel structure. The low-temperature Mössbauer spectra measured at 150 and 5 K indicated that the sample of dried FF contained mainly maghemite nanoparticles [11].

The spectra of the magnetic lignite and magnetic char were almost identical (Figure 5a,b). They had broad magnetic sextets and doublets. The iron-bearing nanoparticles with broad size distributions thus affected the shape of the Mössbauer spectra. Hyperfine parameters (Table 3) of the doublet and sextet components indicated that the Fe^3+^ oxidation state occurred in both samples. The intensity of the superparamagnetic doublet indicated the presence of nanoscale maghemite particles (note that the bulk form of the maghemite was a sextet). The superposition of a superparamagnetic doublet and a broad sextet was due to relaxation near the blocking temperature of the maghemite. 

Iron-oxide surfaces play a major role during sorption, especially when anionic species are removed from aqueous solutions. We studied the oxidation state of the iron on the surfaces of the composites using XPS. The two mechanochemically synthesised sorbents and the initial sample of dried FF were thus examined (Figure 6). The dried FF contained a large amount of iron, but XPS unfortunately recorded a low signal-to-noise intensity, caused by the presence of an oleic acid/surfactant coating layer (note that the probing of the outermost sample surface by XPS was as deep as 10 nm [29]). FF is a homogeneous colloidal suspension, and each nanoparticle must be coated to inhibit clumping. Wilson and Langell [30] observed a similar effect, where magnetite nanoparticles capped with oleylamine/oleic acid were also studied by XPS. The quantitative analysis of concentration could not be correctly evaluated because the low-intensity spectra in the Fe 2p region were attenuated, so only qualitative structure was evaluated. XPS thus indicated the presence of Fe^3+^ ions in the sample of dried FF, which could be assigned to maghemite.

The signal-to-noise intensities, though, were better for the prepared magnetic adsorbents and differed notably between the magnetic composites (magnetic char and magnetic lignite) (Figure 6). The coated iron-oxide particles reacted with the carbon matrices of the char and or lignite during the high-energy milling. The sample of magnetic lignite contained Fe^3+^ in tetrahedral and octahedral positions typical for maghemite. The XPS spectrum was broader for the magnetic char than the magnetic lignite (see mainly the Fe2p_3/2_ region). Magnetic char reflects except of Fe^3+^ also Fe^2+^ ions and metallic iron (Fe^0^), probably due to better reduction conditions during milling, which was caused by the char sample (the char sample contained more carbon than the lignite sample and initiated better reduction conditions). The Fe^2+^ ion could be assigned to Fe_3_O_4_ (magnetite) or FeO. Finally, the presence of reduced iron species (Fe^2+^ and Fe^0^) on the surface of the magnetic char was crucial for the sorption of arsenic anions. Overall, Mössbauer spectroscopy indicated that the two magnetic composites were almost identical. XPS, as a surface-sensitive method, nevertheless identified notable differences between the two magnetic carbon composites.

### 3.4. Effect of pH on Sorption Properties (Cd(II) and As(V))

The study of the influence of pH on the sorption capacity of magnetic sorbents for individual ions is important for sorption. As(V) anions and cationic Cd(II) are present in aqueous solutions in different forms depending on the pH of the solution. pH can also affect the degree of ionisation of functional groups, surface charge of the sorbent, and the interaction between the metal ions and the sorbent [32]. We thus studied the effect of pH on the sorption properties of various types of sorbent (precursors, i.e., char and lignite, and their products) (Figure 7). Only the sample of magnetic char had an excellent affinity to arsenic ions throughout the pH range, consistent with the ZP results and the XPS measurements discussed above. The amounts of adsorbed As were highest within a pH range of 3–4 (Figure 7a), where the sorption capacity was as high as 15 mg/g. Other adsorbents removed only negligible amounts of As(V) (*q*_e_ = 0–2 mg/g), so the subsequent adsorption analysis for arsenic removal (equilibrium adsorption isotherms) used only the magnetic char.

The sorption capacity was quite different for cadmium cations. Coal char was not an effective sorbent, with a sorption capacity <2 mg/g throughout the pH range. Increasing pH led to a linear uptake when other sorbents (lignite, magnetic lignite, and magnetic char) were used (Figure 7b). The char sample had negligible Cd(II) sorption, but the mechanically synthesised magnetic char had a much better Cd(II) sorption affinity, probably due to the larger surface area (S_BET_ = 10.6 m^2^/g). The uptakes were highest and nearly equal for the lignite and magnetic lignite (two-fold higher than for magnetic char at pH 6.5). 

The influence of pH on iron leaching from the prepared sorbents was also studied. All samples were stable in solutions with pHs ≥ 3.8 (Figure 7c). Acidic pH caused the leaching of iron from the magnetic sorbents, especially for the magnetic char, probably due to the larger surface area (better access of H_3_O^+^ molecules to the sorbent pores), the different phase composition of the iron-bearing components on the surface (see XPS data), and the higher total iron concentration in the magnetic composite. Overall, the pH sorption tests indicated that both magnetic sorbents prepared by high-energy ball milling were selective for toxic-metal sorption, either in cationic or anionic forms. 

### 3.5. Adsorption Isotherms

Adsorption isotherms are mathematical models used for the study of sorption mechanisms and affinities between adsorbents (with defined surface properties) and adsorption molecules. Two basic models (Langmuir and Freundlich) were applied for simulating adsorption isotherms for the two magnetic sorbents. Data for adsorption equilibria for As(V) and Cd(II) calculated from linearised Langmuir and Freundlich equations are listed in Table 4. 

Specifically, arsenic (V) uptake was tested only for the magnetic char (Figure 8a), because studies of pH adsorption have found that magnetic lignite is ineffective. As(V) uptake by the magnetic char increased to a concentration of 200 mg As(V)/L and reached equilibrium. 

The Langmuir model described the adsorption better than the Freundlich model, and the theoretical maximum sorption capacity was calculated as 19.9 mg/g (Table 4). This value was much higher than for other similar magnetic sorbents synthesised previously (Table 5). Only a few studies have reported maximum sorption capacities >20 mg/g. Shen et al. [33] demonstrated that the monolayer adsorption capacity of As(V) onto a mesoporous carbon aerogel adsorbent was 56.2 mg/g at pH 7.0. Relatively high maximum sorption capacities were also recorded for commercial adsorbents such as Adsorbsia (Dow Chemical Company (Launch: 2005, Midland, MI, USA), 12–15 mg/g) and ArsenX (SolmeteX, Inc. (Launch: 2004, Northborough, MA, USA), 38 mg/g) [34]. Our previous study [11] demonstrated that a magnetic carbon composite prepared by microwave pyrolysis from agricultural-waste biomass and FF was an effective sorbent of As(V) (*Q*_m_ = 25.6 mg/g). 

The dominant inorganic forms of arsenic in water are As(III) and As(V) [35], where As(III) is more toxic. The As(V) oxyanion is prevalent in surface water, and As(III) is prevalent in groundwater (under anaerobic conditions) [36]. Redox potential (Eh) and pH are the most important factors controlling As speciation [36,37]. The As(V) oxyanion (pKa = 2.22, 6.98, 11.53) dissociates in water as H_3_AsO_4_^0^ (pH < 2.22), H_2_AsO_4_^−^ (pH 2.22–6.98), HAsO_4_^2−^ (pH 6.98–11.53), and AsO_4_^3−^ (pH > 11.53). 

Natural coal and biochars have negative zeta potentials, indicating negative surface charges [2,11,38]. Modification of char precursors by maghemite had a large impact on their negative surface charge. The surface of the composite became more positive, so arsenic presented as an oxyanion (H_2_AsO_4_^−^) was effectively adsorbed. Our results are compatible with previous studies, where the sorption of arsenic on magnetic material was highest at acidic pHs [11,39,40]. A repulsive electrostatic effect was more common in the alkaline environment. The mechanism of the sorption of arsenic on iron oxides has also been studied using the technique of Extended X-ray Absorption Fine Structure [41,42,43,44]. Bidentate binuclear inner-sphere complexes are the most thermodynamically preferred and formed [45,46]:2S = OH + HAsO_4_^2−^ ↔ S_2_ = HAsO_4_ + 2OH^−^
where S represents a surface binding site.

Interestingly, Carabante et al. [40] studied the adsorption of As (V) on iron oxide nanoparticle films by in situ ATR-FTIR (Attenuated Total Reflection—Fourier Transform Infrared) spectroscopy in D_2_O (heavy water) at pD 4 and 8.5 and found that inner-sphere complexes (Fe-O_4_H_2_As) formed at pD 4 were more stable than complexes formed at pD 8.5 due to outer-sphere complexation. 

Both magnetic sorbents had good sorption properties for removing Cd(II) (Figure 9a). The maximum adsorption capacity was higher for the magnetic lignite (*Q*_m_ = 70.4 mg/g) than the magnetic char (*Q*_m_ = 58.8 mg/g) (Table 4). Metal uptake increased to 100 mg Cd(II)/L for both magnetic adsorbents and then reached equilibrium. The better sorption properties of the magnetic lignite towards cadmium cations corresponded to the surface charge of the adsorbent (ZP was lower for magnetic lignite than magnetic char (Figure 3)). *Q*_m_ for both were comparable to results from sorption experiments with various magnetic composites (Table 5).

Both models reproduced sorption processes very well, with *R*^2^ > 0.98 (linearised Langmuir isotherm) and >0.93 (linearised Freundlich isotherm). The experimental data, however, were more strongly correlated with the Langmuir model in all cases, indicating that adsorption on monolayers on surfaces containing a finite number of identical sorption sites and interaction between molecules were negligible. The specific surface area and the surface charge and chemistry notably influenced the adsorption capacity of the adsorbents. Our study demonstrated that the magnetic lignite was a convenient adsorbent for Cd(II) removal, despite its small specific surface area. Macro/mesoporous magnetic char, though, had excellent affinity for As(V) removal.

To sum up, non-conventional high-energy ball milling of lignite/char with FF at a laboratory scale produced effective magnetic sorbents for removing toxic metals. Our results indicated that the adsorbents were stable at pHs ≥ 3.8 (Figure 7c). We therefore assumed that the synthesised magnetic sorbents could be used in freshwater systems. Moreover, both sorbents also had stable magnetic properties (volumetric magnetic susceptibility did not change during adsorption). Other stability tests such as the influence of ionic strength, temperature, and time (testing of degree of oxidation), however, must be provided in the future. The large-scale production of magnetic sorbents by high-energy ball milling is another issue. Scaling up the production of sorbents mechanochemically is still a demanding task, due mostly to the high processing cost, especially when long reaction times and closed systems with defined reaction atmospheres are needed.

## 4. Conclusions

Effective iron-based natural biosorbents with strong magnetic properties can be prepared by a simple one-step mechanochemical procedure: high-energy ball milling of lignite/char with FF. The synthesised magnetic sorbents were effective materials for removing cadmium cations (*Q*_m_ (magnetic lignite) = 70.4 mg/g; *Q*_m_ (magnetic char) = 58.8 mg/g). Magnetic char was also an excellent sorbent of arsenic oxyanions (*Q*_m_ (magnetic char) = 19.9 mg/g) in aqueous solutions. Removal by magnetic filtration during water purification is a great advantage of magnetic sorbents.

## Figures and Tables

**Figure 1 nanomaterials-09-00100-f001:**
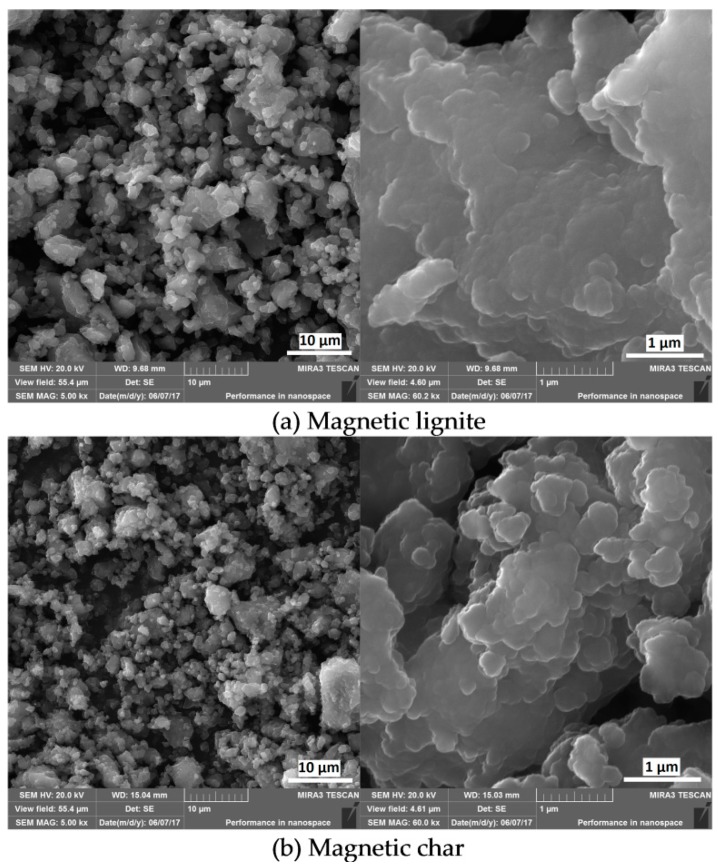
Field-emission scanning electron micrographs of the (**a**) magnetic lignite and (**b**) magnetic char.

**Figure 2 nanomaterials-09-00100-f002:**
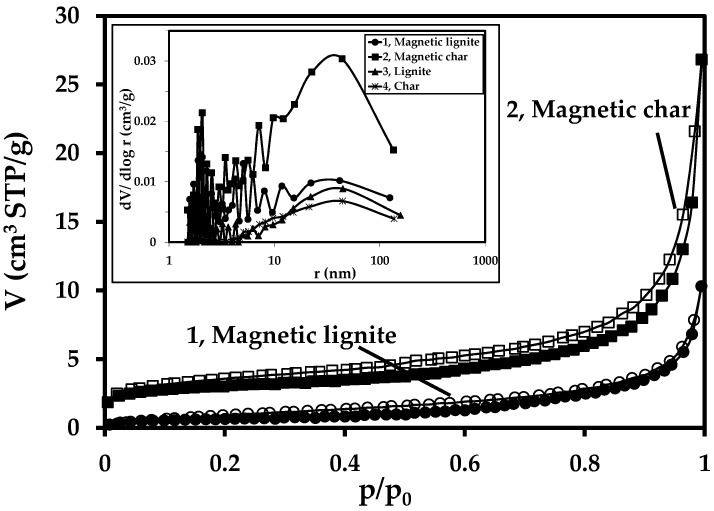
Nitrogen adsorption/desorption isotherm of magnetic lignite (1) and magnetic char (2). Solid datapoints in the isotherm correspond to the data from adsorption isotherm, and open datapoints correspond to the data from the desorption isotherm. Top left inset: pore-size distribution for magnetic lignite (1) and magnetic char (2) and their precursors, lignite (3) and char (4), derived from the desorption isotherms. STP, standard temperature and pressure (t = 0 °C and p = 101.325 kPa).

**Figure 3 nanomaterials-09-00100-f003:**
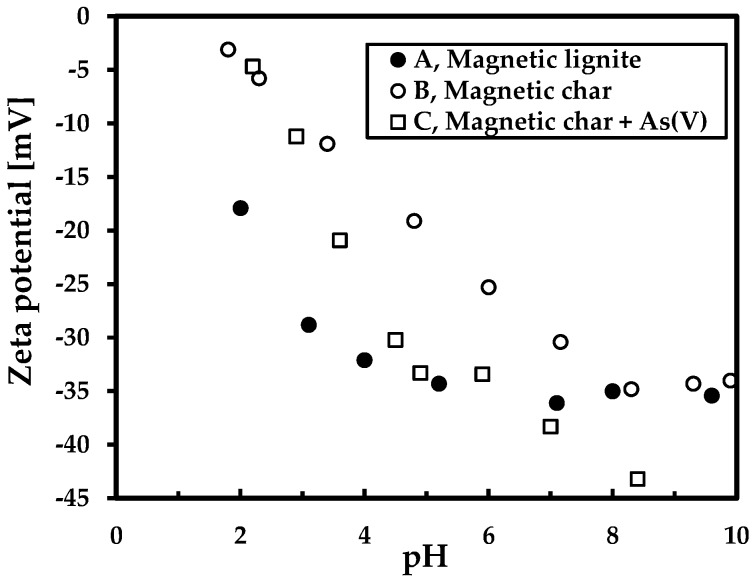
Zeta potential of the sorbents at various pHs in 0.1 M NaNO_3_ (sample C: c (AsV) = 100 ppm).

**Figure 4 nanomaterials-09-00100-f004:**
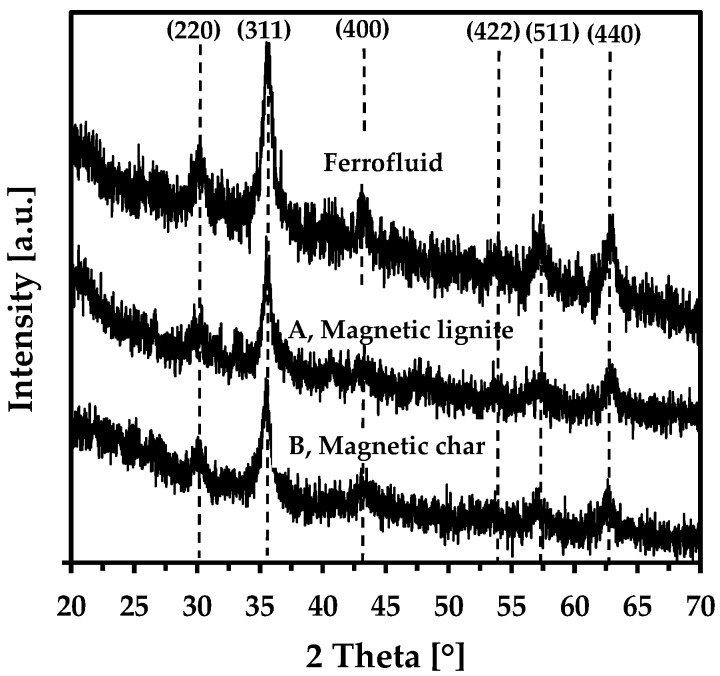
X-ray powder diffraction of the dried ferrofluid, magnetic lignite, and magnetic char (Miller indices for maghemite/magnetite are also indicated).

**Figure 5 nanomaterials-09-00100-f005:**
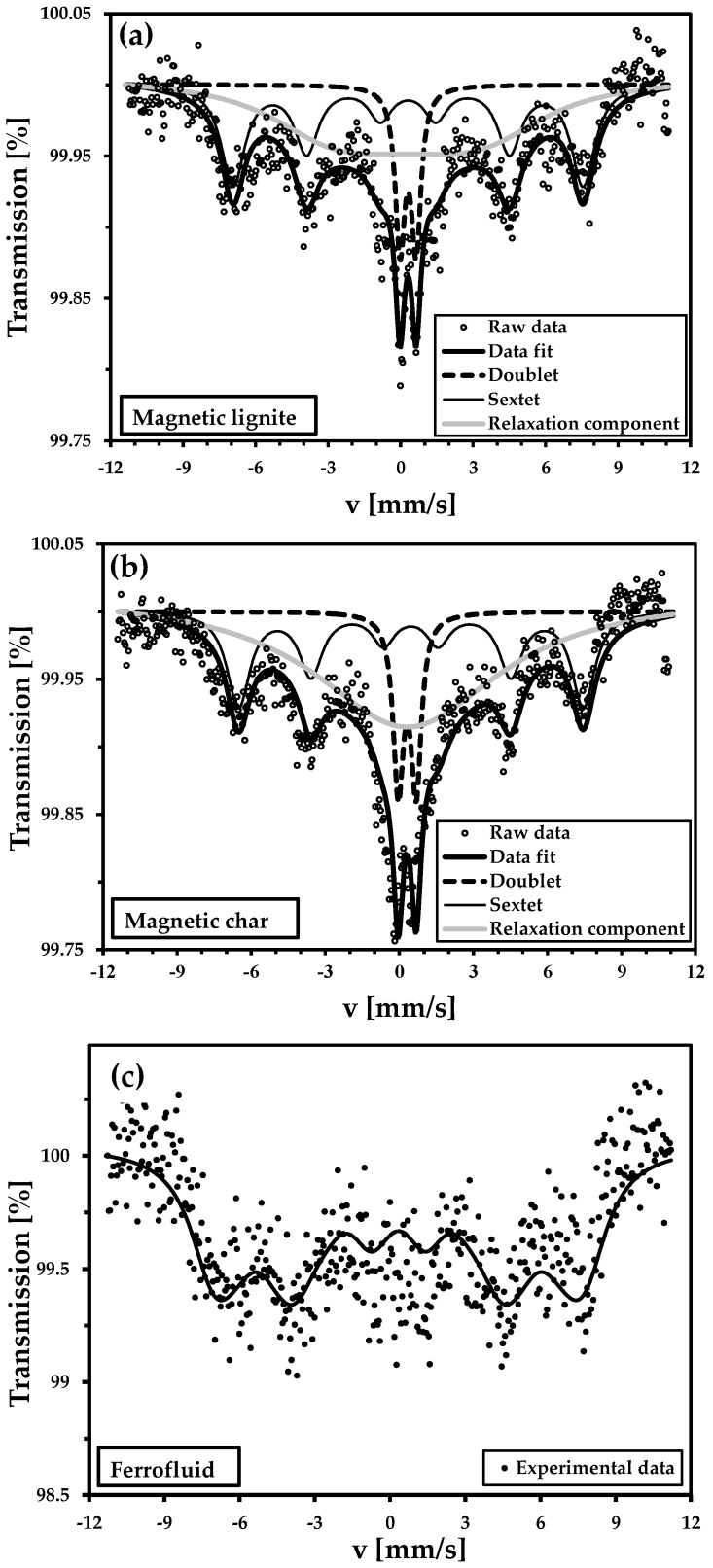
Room-temperature ^57^Fe Mössbauer spectra of the magnetic lignite (**a**), magnetic char (**b**), and dried ferrofluid (**c**).

**Figure 6 nanomaterials-09-00100-f006:**
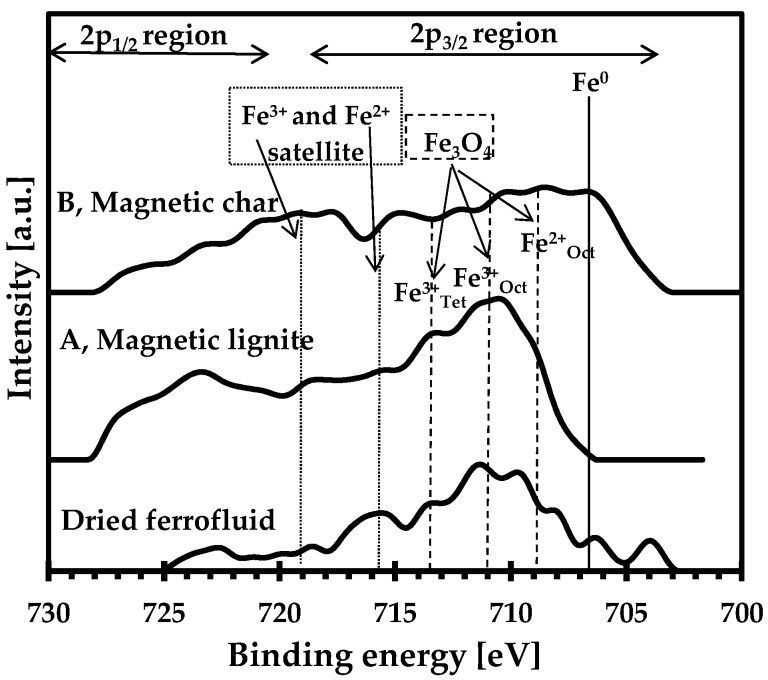
Background-subtracted Fe 2p XPS spectra of the sorbents (magnetic lignite and magnetic char) and dried ferrofluid. The multiple peaks of Fe^3+^ and Fe^2+^ in the Fe 2p_3/2_ region were labelled in the graph as described by [30,31].

**Figure 7 nanomaterials-09-00100-f007:**
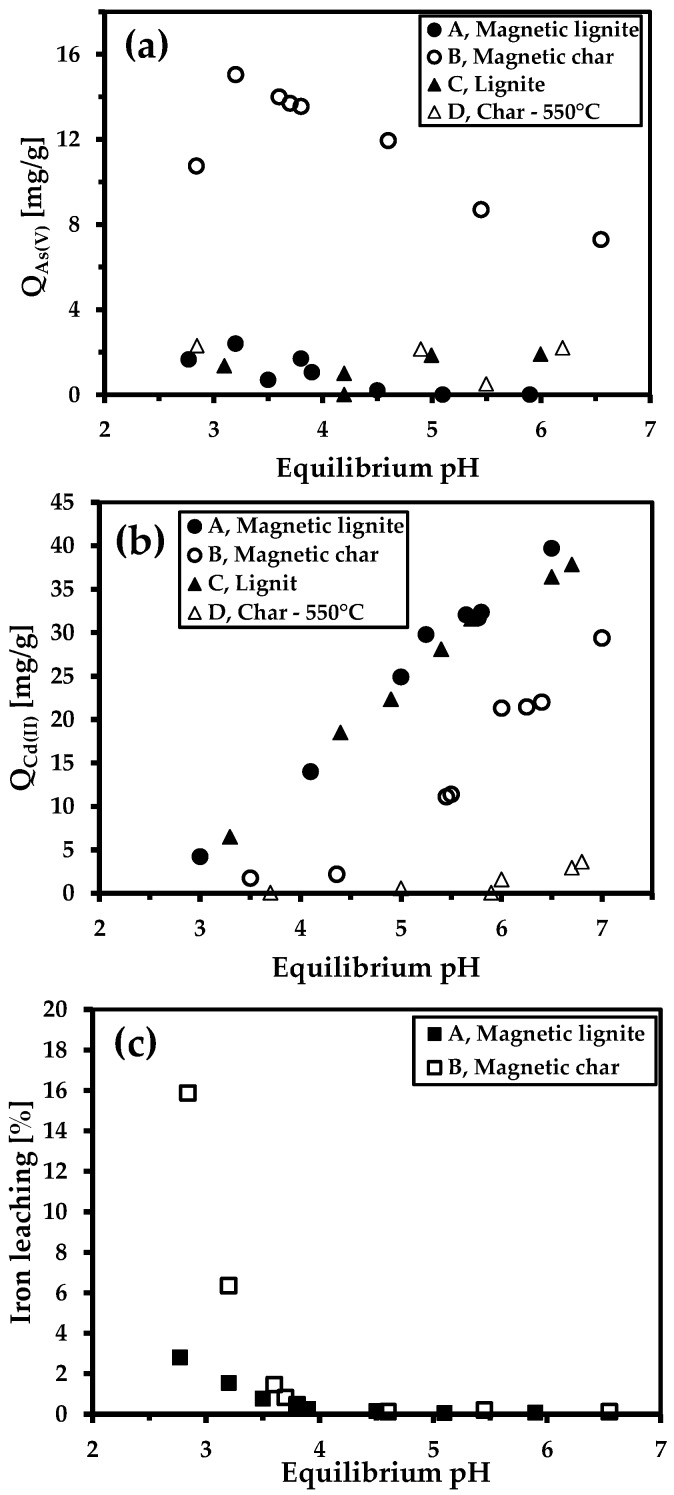
Sorption capacities of the sorbents towards As(V) (**a**) and Cd(II) (**b**) at various pHs. Iron leaching during the sorption of As(V) at various pHs (**c**). Conditions: batch-type system; c(sorbent) = 2 g/L; c_0_(As(V)) = 98.6 mg/L; c_0_(Cd(II)) = 92 mg/L; room temperature; 24 h.

**Figure 8 nanomaterials-09-00100-f008:**
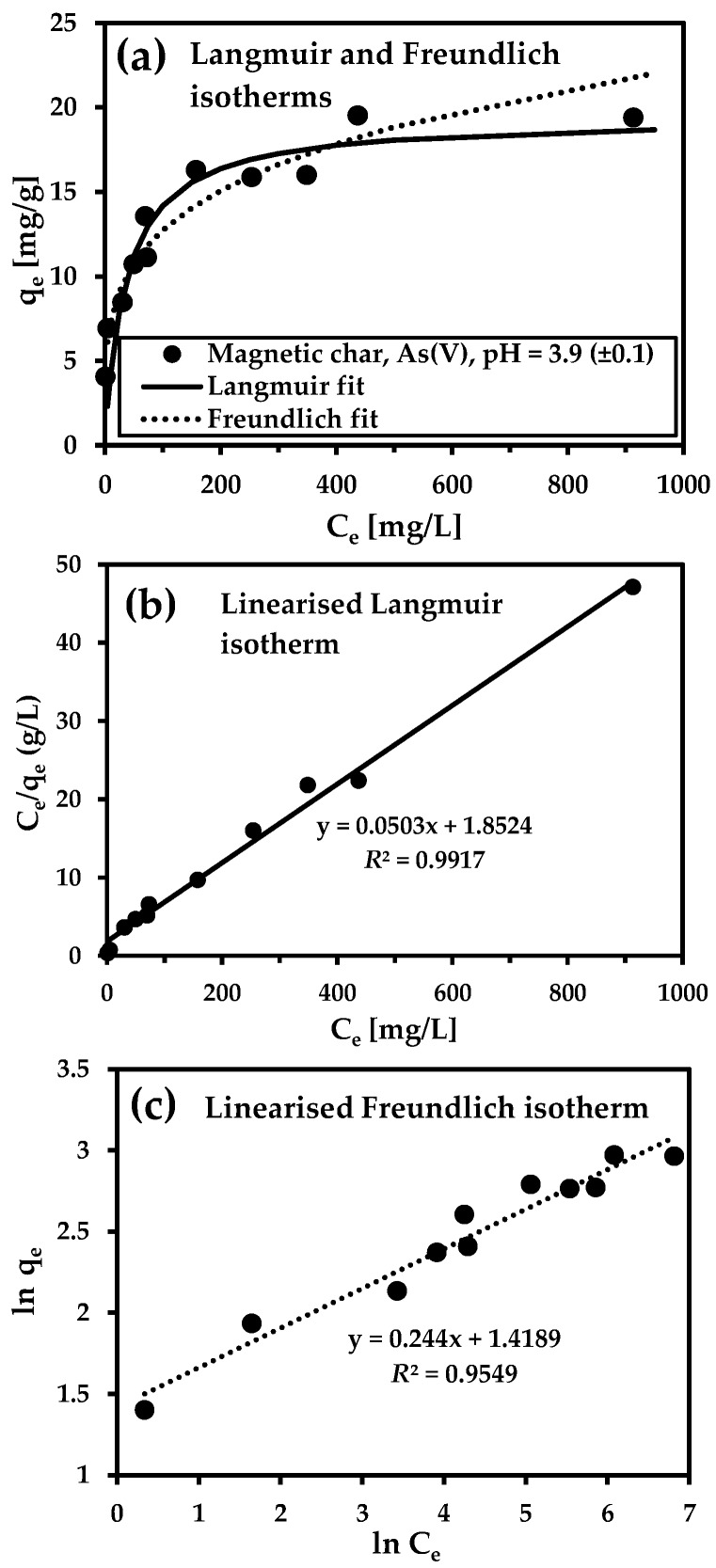
Langmuir and Freundlich adsorption isotherms for As(V) (**a**). The adsorption isotherms were calculated using the linearised forms of the Langmuir (**b**) and Freundlich (**c**) equations. Conditions: sample, magnetic char; batch-type system; sorbent concentration, 2 g/L; initial metal concentration, 10–1000 mg/L; room temperature; 24 h).

**Figure 9 nanomaterials-09-00100-f009:**
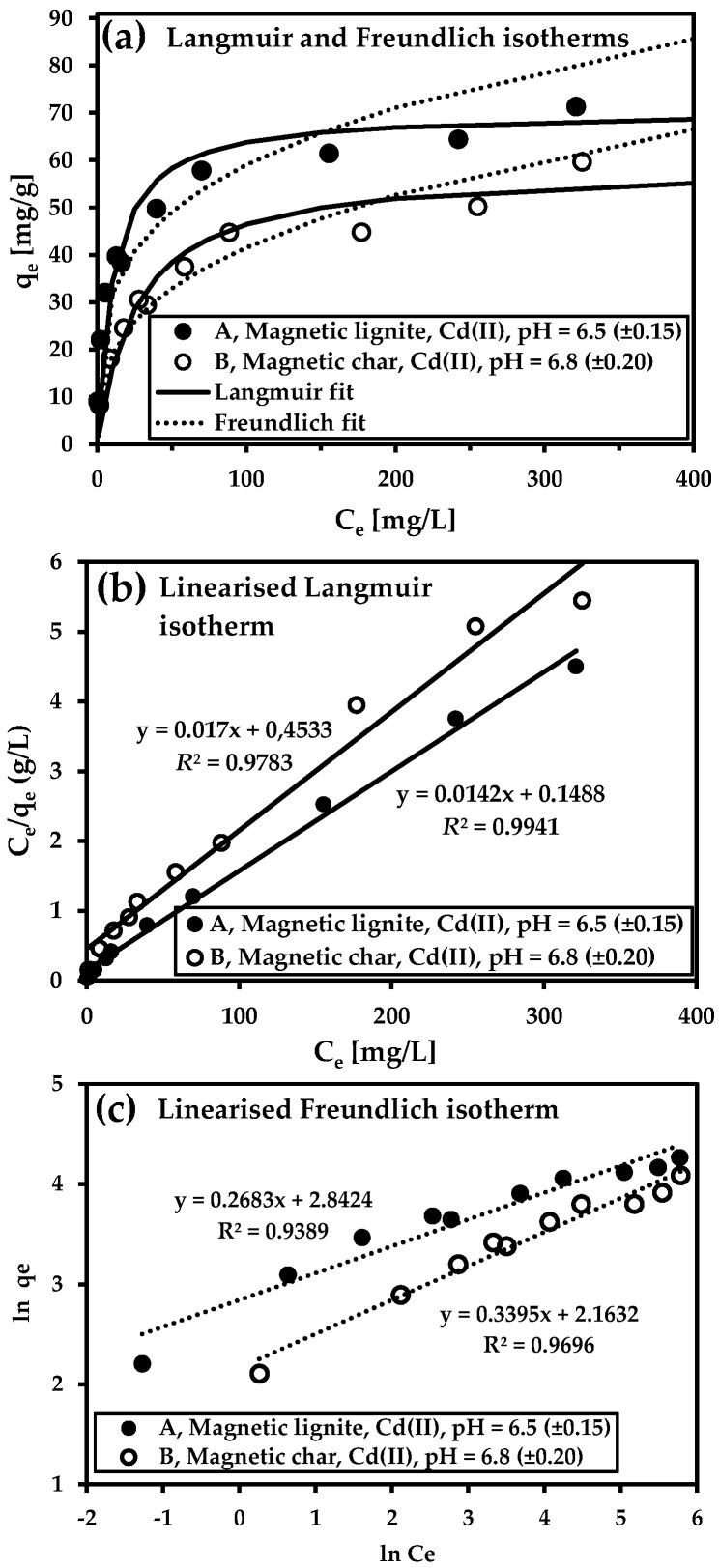
Langmuir and Freundlich adsorption isotherms for Cd(II) (**a**). The adsorption isotherms were calculated using the linearised forms of the Langmuir (**b**) and Freundlich (**c**) models. Conditions: samples, magnetic char and magnetic lignite; batch-type system; sorbent concentration, 2 g/L; initial metal concentration, 10–464 mg/L; room temperature; 24 h).

**Table 1 nanomaterials-09-00100-t001:** Elemental (CHNS) analysis, ash content, total iron content, and volumetric magnetic susceptibility (κ) of the initial samples (lignite and char) and the samples after high-energy ball milling (magnetic lignite and magnetic char).

Sample	A^d^ (%)	C^d^ (%)	H^d^ (%)	N^d^ (%)	S^d^ (%)	O^d^ (%)	Fe_Total_ (%)	κ (SI Units)
Lignite	13.8	51.9	5.4	0.8	2.3	25.8	-	89 × 10^−6^
Char	21.6	62.9	3.1	1.0	2.4	9.0	-	223 × 10^−6^
Magnetic lignite	-	46.7	5.4	0.7	2.0	-	7.7	343,333 × 10^−6^
Magnetic char	-	55.5	3.5	0.9	2.2	-	8.5	316,897 × 10^−6^

Abbreviations: A, ash; d, dry-weight basis; O^d^, by difference calculated as 100 − (A^d^ + C^d^ + H^d^ + N^d^ + S^d^).

**Table 2 nanomaterials-09-00100-t002:** Textural parameters for the magnetic lignite and magnetic char after high-energy ball milling. Data for the natural lignite sample and the coal char are also presented. Specific surface area (S_BET_) and the C_BET_ constant were calculated from the BET isotherm, total pore volume (V_tot_) was estimated from the adsorbed gas volume at a relative pressure near 1, and the volume of micropores and the external surface (S_ext_) were determined from a t-plot.

Sample	S_BET_ (m^2^/g)	*C* _BET_	V_tot_ (cm^3^/g)	V_micro_ (cm^3^/g)	S_ext_ (m^2^/g)
Lignite	1.0	41.2	0.0099	0	1.0
Char	1.8	−350.5	0.0084	0.0019	0
Magnetic lignite	2.3	69.1	0.0159	0	2.2
Magnetic char	10.6	−343.5	0.0415	0.0024	5.3

**Table 3 nanomaterials-09-00100-t003:** Mössbauer parameters for the magnetic lignite and magnetic char. Mössbauer spectra are presented in Figure 5a,b.

Sample	IS (mm/s)	QS (mm/s)	H (T)	*I* (%)	Component Description
Magnetic lignite	0.31	0.67	-	10.6	Doublet
	0.45	0.001	45.0	33.6	Sextet
	0.31	−0.09	22.1	55.8	Relaxation component
Magnetic char	0.32	0.72	-	9.5	Doublet
	0.47	0.005	43.4	23.6	Sextet
	0.31	0.009	0	66.9	Relaxation component

Abbreviations: H, magnetic field; I, percentage distribution; QS, quadrupole splitting; IS, isomer shift.

**Table 4 nanomaterials-09-00100-t004:** Langmuir and Freundlich model parameters for magnetic char and magnetic lignite.

Sample Description	Langmuir Model			Freundlich Model		
	*Q*_m_ (mg/g)	*b* (L/mg)	*R* ^2^	*K_F_* (L/g)	*n*	*R* ^2^
Magnetic char—As(V), pH 3.9	19.9	0.02715	0.9917	4.13	4.10	0.9549
Magnetic lignite—Cd(II), pH 6.5	70.4	0.09543	0.9941	17.16	2.95	0.9389
Magnetic char—Cd(II), pH 6.8	58.8	0.03750	0.9873	8.70	3.73	0.9696

**Table 5 nanomaterials-09-00100-t005:** Comparison of maximum sorption capacity (*Q*_m_) of arsenic and cadmium for the magnetic carbons, iron oxides, and commercial adsorbents.

**Sorption of Arsenic (V)**	***Q*_m_ (mg/g)**	**Reference**
Adsorbsia, Dow Water Solutions (Launch: 2005, USA)	12–15	[34]
ArsenX, SolrneteX, Inc. (Launch: 2004, USA)	38	[34]
Magnetic biochar (wheat straw)	25.6	[11]
Iron-impregnated biochar	2.2	[47]
Biochar/γ-Fe_2_O_3_ composite	3.2	[48]
Iron-oxide amended rice-husk char	1.5	[49]
γ-Fe_2_O_3_	4.6	[50]
Fe_3_O_4_	4.7	[51]
Flowerlike γ-Fe_2_O_3_	4.8	[51]
*α*-Fe_2_O_3_	5.3	[51]
Iron-modified activated carbon	1.9–6.6	[52]
Magnetic mesoporous carbon aerogel	56.2	[33]
Magnetic chitosan biochar	17.9	[53]
Iron oxide/activated carbon	20.2	[54]
Hematite-modified biochar	0.43	[15]
Magnetic char	19.9	Present study
**Sorption of Cadmium (II)**		
Magnetic biochar	15.0	[55]
Magnetic biochar	62.5	[56]
Chitosan-modified magnetic biochar	105.3	[57]
Magnetic activated carbon	49.8	[58]
Magnetic oak-bark biochar	8.3	[59]
Magnetic biochar (coconut shell)	3.9	[60]
Magnetic biochar (Douglas fir biochar)	11.3	[61]
Magnetic biochar (mangosteen peels)	45.7	[62]
Magnetic char	58.8	Present study
Magnetic lignite	70.4	Present study

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
