# Peer review of "Mechanochemically Synthesised Coal-Based Magnetic Carbon Composites for Removing As(V) and Cd(II) from Aqueous Solutions"

_nanomaterials, 2019, doi:10.3390/nano9010100_

Reviewer 1 Report

Although research design and materials are rather standard, quality of data organization is high enough to get publication in scientific research journals. Especially, inclusion of Mossbauer spectra is impressive. Therefore, I basically recommend publication of this work in Nanomaterials. Rather than scientific contents, I have several revision requests on general matters.

1) Reference selection is not well-updated and general enough. Recent references to the initial general descriptions in Introduction have to be strengthened where recent papers on detection and removal of toxic materials from water had better be added (for example, Bulletin of the Chemical Society of Japan 2017, 90, 678-683, Journal of Hazardous Materials 2018, 342, 732-740).

2) Please provide clearer scale bars to images in Figure 1.

3) In Figure 3, horizontal axis (pH) had better come to the bottom (not top).

4) XPS data are rather ambiguous. If possible, peak splitting and more detailed analyses had better be done.

Author Response

Response to comments of Reviewer 1:

Although research design and materials are rather standard, quality of data organization is high enough to get publication in scientific research journals. Especially, inclusion of Mossbauer spectra is impressive. Therefore, I basically recommend publication of this work in Nanomaterials. Rather than scientific contents, I have several revision requests on general matters.

 1) Reference selection is not well-updated and general enough. Recent references to the initial general descriptions in Introduction have to be strengthened where recent papers on detection and removal of toxic materials from water had better be added (for example, Bulletin of the Chemical Society of Japan 2017, 90, 678-683, Journal of Hazardous Materials 2018, 342, 732-740).

Response: The new references were added to the Introduction section. See page 1-2 (lines 42-50). 

2) Please provide clearer scale bars to images in Figure 1.

Response: Figure 1 was edited.

3) In Figure 3, horizontal axis (pH) had better come to the bottom (not top).

Response: Figure 3 was edited according to your comment.

4) XPS data are rather ambiguous. If possible, peak splitting and more detailed analyses had better be done.

Response: We tried to fit the spectra, but the statistics (XPS signal-to-noise) obtained by XPS measurement of FF as well as our composites are unfortunately really poor (note that the probing of the outermost sample surface by XPS was as deep as 10 nm (Haasch, R.T. X-ray photoelectron spectroscopy (XPS) and auger electron spectroscopy (AES). In Practical materials characterization, Springer: 2014; pp 93-132.)). Our FF is coated with oleic acid, and iron-bearing phases are incorporated into the carbon matrix of the composites. The fitting of individual iron-bearing components is therefore theoretically possible but could not be correctly evaluated. Additionally, fitted sub-spectra of individual Fe-phases will overlap and be difficult to recognize in the figure. Finally, we decided to highlight and compare the main Fe peaks of our inspected samples in 2p regions. The differences between FF and the synthesized composites are clearly visible.

Reviewer 2 Report

This work is well carried out

However, for any application-oriented work the stability of the material in the targeted system, fresh water, in this case, should be discussed.

Are the lignite/char are chemically bound with the ferrite? 

Is there any leaching of the material into the water during adsorption?

What is the mechanism of adsorption? Why is lignite ineffective for As(V)?

Both the application-oriented discussion (stability and selectivity in the real system) and the material characterization discussion (mechanism, if something is exchanged with the target ion, etc), need to be improved.

Author Response

Response to Comments of Reviewer 2:

This work is well carried out

However, for any application-oriented work the stability of the material in the targeted system, fresh water, in this case, should be discussed.

Response: A discussion of the stability of the sorbents was incorporated into the Results and Discussion section. See page 16 (lines 452-458). 

Are the lignite/char are chemically bound with the ferrite? 

Response: There may be no chemical bond between carbon and iron, because Mossbauer spectroscopy is a sensitive method and would detect new phases such as carbides (non-stoichiometric Fe3C). Maghemite nanoparticles were recorded in the Mossbauer spectra.  

Is there any leaching of the material into the water during adsorption?

Response: Magnetic sorbents were stable in solutions with pHs ≥3.8 (Figure 7c). The acidic environment caused the leaching of iron from the composites.

What is the mechanism of adsorption? Why is lignite ineffective for As(V)?

Response: New text focusing on the mechanism of adsorption was added to the Results and Discussion section. See page 14 (lines 410-429).

Both the application-oriented discussion (stability and selectivity in the real system) and the material characterization discussion (mechanism, if something is exchanged with the target ion, etc), need to be improved.

Response: Thank you for this comment. A discussion of the sorption mechanism and the stability (application-oriented discussion) of the sorbents was added to the Results and Discussion section.

Reviewer 3 Report

This is a well written paper contrasting two carbon based absorbents for their ability to remove a toxic heavy metal (Cd) and a toxic anionic waste (As(V)).  Both adsorbents provide potential new sinks for these heavy metals.  I detected little in the way of technical error, and the paper is reflective of many similar papers of the same theme.  My only real criticisms are general to the genre of research:

. Why is it important for the adsorbent to have magnetic properties?

. Are you trading one toxic problem for another (i.e. toxic heavy metal containing carbon composites).

. Why is high powered ball milling considered non-conventional - there are many examples of this in the literature.

.  Why is the Smulochowski equation still used when it is so well known that it fails in the colloidal region?

.  Given that there is so much ash, for example, how much does this contribute to the adsorption.  What are the surface groups responsible for adsorption?

. How does the pH dependence compare with literature, and in particular the trend for adsorption to match the pKa of the surface sites (see Eldridge, et al., The role of metal ion-ligand interactions during divalent metal ion adsorption. Journal of Colloid and Interface Science 2015, 454 (Supplement C), 20-26

Author Response

Response to Comments of Reviewer 3:

This is a well written paper contrasting two carbon based absorbents for their ability to remove a toxic heavy metal (Cd) and a toxic anionic waste (As(V)). Both adsorbents provide potential new sinks for these heavy metals. I detected little in the way of technical error, and the paper is reflective of many similar papers of the same theme. My only real criticisms are general to the genre of research:

. Why is it important for the adsorbent to have magnetic properties?

Response: We have explained it thoroughly in the Introduction section. See page 2 (lines 51-70).

. Are you trading one toxic problem for another (i.e. toxic heavy metal containing carbon composites).

Response: Cadmium and arsenic are highly toxic pollutants. According to the US Environmental Protection Agency, the maximum contaminant level for cadmium in drinking water is 0.005 mg/L and the maximum contaminant level for arsenic is 0.01 mg/L. Both metals cause serious medical complications such as cancer. The acceptable value of iron in drinking water is 0.3 mg/L. Iron overload may lead to debilitating and life-threatening problems. A high concentration of iron was detected mainly in industrial wastewater (Dahlan et al., Sustainable Environment Research 2013, 23, 41-4.). Our magnetic sorbents are stable in aqueous environments with pHs ≥3.8. The acidic environment caused the leaching of iron from the composites (Figure 7c). Prepared magnetic sorbents can be easily removed together with adsorbed pollutants from aqueous environments by an external magnetic field.

New text was included to the Introduction section. See page 3 (lines 104-107).

. Why is high powered ball milling considered non-conventional - there are many examples of this in the literature.

Response: Yes, there are many examples in the literature focused on the mechanochemical synthesis of new materials (e.g. nanomaterials) with different properties. However the high-energy ball milling is still not adapted at the industrial scale as conventional methods (precipitation, pyrolysis). The synthesis of effective sorbents (magnetic sorbents) by high-energy ball milling is non-conventional. 

We have explained it in Results and Discussion section. Page 16-17 (lines 458-461).      

.  Why is the Smulochowski equation still used when it is so well known that it fails in the colloidal region?

Response: According to the manual for the equipment (Zeta sizer), Henry’s equation is used to convert the data for measured electrophoretic mobility into the zeta potential:

Ue = 2εz ƒ(Ka)
  3 η

where f(Ka) is Henry’s function. For particles larger than about 0.2 microns, f(Ka) becomes 1.5, which is referred to as the Smoluchowski approximation. In our case, particles are >200 nm. Maghemite nanoparticles are incorporated into the carbon matrix.                                            

Particle-size analysis of the magnetic char using a Nanophox particle-size analyzer (Sympatec, Germany).

Section 2.9. Zeta potential (ZP) was edited. Page 4 (lines 175-179).

.  Given that there is so much ash, for example, how much does this contribute to the adsorption.  What are the surface groups responsible for adsorption?

Response: Ash naturally occurs in natural carbonaceous materials such as plant biomass and coal. Ash is primarily made of compounds of silica, aluminum, iron, and other substances. It can play a negative or positive role during adsorption. The amount of ash usually increases after pyrolysis, because the oxide functional group is removed from the initial material. We also tested our precursors (lignite and char) as sorbents of As(V) and Cd(II) (Figure 7a, b). Lignite and char did not effectively remove arsenic, but for cadmium, natural lignite with lower ash content is an effective sorbent. The effect of the ash content of natural lignocellulosic material could be interesting for our future studies.

The mechanism of adsorption was described in Results and Discussion section. See page 14 (lines 410-429).

. How does the pH dependence compare with literature, and in particular the trend for adsorption to match the pKa of the surface sites (see Eldridge, et al., The role of metal ion-ligand interactions during divalent metal ion adsorption. Journal of Colloid and Interface Science 2015, 454 (Supplement C), 20-26

Response: Thank you for this comment. A discussion of the sorption mechanism has been included in the Results and Discussion section (page 14 (lines 410-429)).

Reviewer 4 Report

Although the presentation of this paper is good, my sense is that it cano be published. Authors successfully achieved the synthesis of a material which can adsorb As(V) and Cd(II). But screening literature and based on table of authors, there is a huge number of papers with those ions removed from other sorbents in impressively higher capacity.

Author Response

Response to Comments of Reviewer 4:

Although the presentation of this paper is good, my sense is that it cano be published. Authors successfully achieved the synthesis of a material which can adsorb As(V) and Cd(II). But screening literature and based on table of authors, there is a huge number of papers with those ions removed from other sorbents in impressively higher capacity.

Response: We consider our research to be relevant. The topic of the removal of carcinogenic pollutants such as arsenic and cadmium is very important. Rapid industrialisation has produced hazardous secondary products on a large scale, polluting water especially in developing countries. Several methods including adsorption are developed for eliminating toxic substances either in organic or inorganic (heavy metals) forms. Many papers with similar topics devoted to the synthesis of cheap and effective adsorbents have therefore been published.

Round  2

Reviewer 2 Report

improved well.

Author Response

Thank you for the revision.

Reviewer 4 Report

I recommend the rejection due to lack of REAL novelty.

Author Response

Thank you for the revision.